# First Identification and Investigation of piRNAs in the Larval Gut of the Asian Honeybee, *Apis cerana*

**DOI:** 10.3390/insects14010016

**Published:** 2022-12-23

**Authors:** Qi Long, Ming-Hui Sun, Xiao-Xue Fan, Zong-Bing Cai, Kai-Yao Zhang, Si-Yi Wang, Jia-Xin Zhang, Xiao-Yu Gu, Yu-Xuan Song, Da-Fu Chen, Zhong-Min Fu, Rui Guo, Qing-Sheng Niu

**Affiliations:** 1College of Animal Sciences (College of Bee Science), Fujian Agriculture and Forestry University, Fuzhou 350002, China; 2Jilin Institute of Apicultural Research, Jilin 132000, China; 3Apitherapy Research Institute, Fujian Agriculture and Forestry University, Fuzhou 350002, China

**Keywords:** honeybee, *Apis cerana*, *Apis cerana cerana*, piRNA

## Abstract

**Simple Summary:**

Piwi-interacting RNAs (piRNAs) exert crucial functions in maintaining the genomic stability and modulating gene expression and biological processes. However, study on piRNAs in *Apis cerana* is still lacking until now. Here, systematic characterization of piRNAs in the larval guts of *Apis cerana cerana* was for the first time conducted, followed by in-depth investigation of the regulatory parts of differentially expressed piRNAs (DEpiRNAs) in the gut developmente. A total of 621 piRNAs were identified in the larval guts, with similar length distribution and first base bias to piRNAs discovered in other insects and mammals. In addition, ten and 22 DEpiRNAs were respectively detected in the Ac4 vs. Ac5 and Ac5 vs. Ac6 comparison groups, with piR-ace-748815, piR-ace-512574, piR-ace-716466, and piR-ace-828146 targeting the highest number of mRNAs. Further, these targets were engaged in several vital pathways relevant to growth and development, such as the Jak/STAT, TGF-β, and Wnt signaling pathways. Our findings provide a new insight into the development of the *A. cerana* larval gut and a basis for illuminating the piRNA-regulated mechanism underlying the gut development.

**Abstract:**

Piwi-interacting RNAs (piRNAs), a class of small non-coding RNAs (ncRNAs), play pivotal roles in maintaining the genomic stability and modulating biological processes such as growth and development via the regulation of gene expression. However, the piRNAs in the Asian honeybee (*Apis cerana*) are still largely unknown at present. In this current work, on the basis of previously gained high-quality small RNA-seq datasets, piRNAs in the larval gut of *Apis cerana cerana*, the nominated species of *A. cerana*, were identified for the first time, followed by an in-depth investigation of the regulatory roles of differentially expressed piRNAs (DEpiRNAs) in the developmental process of the *A. c. cerana*. Here, a total of 621 piRNAs were identified in *A. c. cerana* larval guts, among which 499 piRNAs were shared by 4-(Ac4 group), 5-(Ac5 group), and 6-day-old (Ac6 group) larval guts, while the numbers of unique ones equaled 79, 37, and 11, respectively. The piRNAs in each group ranged from 24 nucleotides (nt) to 33 nt in length, and the first base of the piRNAs had a cytosine (C) bias. Additionally, five up-regulated and five down-regulated piRNAs were identified in the Ac4 vs. Ac5 comparison group, nine of which could target 9011 mRNAs; these targets were involved in 41 GO terms and 137 pathways. Comparatively, 22 up-regulated piRNAs were detected in the Ac5 vs. Ac6 comparison group, 21 of which could target 28,969 mRNAs; these targets were engaged in 46 functional terms and 164 pathways. The results suggested an overall alteration of the expression pattern of piRNAs during the developmental process of *A. c. cerana* larvae. The regulatory network analysis showed that piR-ace-748815 and piR-ace-512574 in the Ac4 vs. Ac5 comparison group as well as piR-ace-716466 and piR-ace-828146 in the Ac5 vs. Ac6 comparison group were linked to the highest number of targets. Further investigation indicated that targets of DEpiRNAs in the abovementioned two comparison groups could be annotated to several growth and development-associated pathways, such as the Jak/STAT, TGF-β, and Wnt signaling pathways, indicating the involvement of DEpiRNAs in modulating larval gut development via these crucial pathways. Moreover, the expression trends of six randomly selected DEpiRNAs were verified using a combination of stem-loop RT-PCR and RT-qPCR. These results not only provide a novel insight into the development of the *A. c. cerana* larval gut, but also lay a foundation for uncovering the epigenetic mechanism underlying larval gut development.

## 1. Introduction

Piwi-interacting RNAs (piRNAs), a class of small non-coding RNAs (ncRNAs) with a length range of 23–35 nucleotides (nt), are abundant in germ cells and reproductive tissues [1,2]. They are generated from single-stranded precursor transcripts independently of Dicer and possess 2′-O-methylation at their 3′ end. For piRNAs in *Caenorhabditis elegans* and *Drosophila melanogaster,* the production of complementary piRNAs occurs with an overlap of ten nt and a 5′ nt bias to uracil (U) [2]. PiRNAs were first identified in *D. melanogaster*, and then in mice [1], humans [3], and *C. elegans* [4]. As piRNAs continue to be discovered in various species, the sequences of piRNAs have been found to have more diverse nt bias features, including a 5ʹ nt bias to U, guanine (G) [2], and cytosine (C) [5]. In insects, the limited work has been performed was mainly relevant to a few model species, such as *Drosophila* [3], silkworms [6], and flies [7]. For example, Xiol et al. identified abundant piRNAs in *Bombyx mori* BmN4 cells on the basis of Illumina deep sequencing and discovered that the inhibition of the ATP-dependent Hsp90 activity in silkworm cells resulted in the accumulation of short antisense RNAs in Piwi complexes [8]. However, a study on other insects including honeybees is still in the preliminary stages. A major function of piRNAs is to maintain genome stability by suppressing the activity of transposons [9]. In recent years, with the development of related knowledge and technology, piRNAs have been suggested to modulate gene expression through targeting mRNAs, similar to the action of miRNAs, another class of small RNAs [10]. For example, Manage et al. found that SIMR-1 interacted with PRG-1 in the piRNA pathway to promote the expression of downstream target genes by the mutator complex [11]. Shen et al. discovered that piRNAs in *C. elegans* could target all mRNAs in the germline following miRNA-like pairing rules [12].

*Apis cerana* is wildly reared in China and many other Asian countries, playing an essential role in the ecology and economy through the pollination of wildflowers and crops, as well as in the production of api-products [13]. Previously, our group conducted a series of studies on ncRNAs regulating the larval development and immune defense of *Apis cerana cerana*, the nominated subspecies of *A. cerana*. Feng et al. performed a transcriptome-wide identification study of miRNAs in *A. c. cerana* larval guts followed by validation of the expression and sequence of six miRNAs [14]. Du et al. deciphered the differential expression profiles of lncRNAs in the *A. c. cerana* larval gut in response to *Ascosphaera apis* infection [15]. Chen et al. analyzed the expression patterns of circRNAs in the larval gut of *A. c. cerana* responding to *A. apis* invasion and unraveled the putative function of differentially expressed circRNAs (DEcircRNAs) in the host immune response [16]. Compared with *Drosophila* and *Aedes albopictus*, the studies on honeybee piRNAs are still limited. Liao et al. identified two PIWI genes, *Am-aub* and *Am-ago3*, for the first time in worker, drone, and queen bees of *Apis mellifera*, and the expression of these two genes showed a significant sex bias, confirming that piRNAs are truly expressed in the Western honeybee, indicating that piRNAs may have a potential association with honeybee reproduction under the influence of nutritional factors, as well as in bee sex determination [17]. Based on RT-qPCR detection, Wang et al. identified piRNA clusters among drone, worker, and queen bees, and found that the expression level of piRNAs in reproductive individuals of *A. mellifera* was greater than that in sterile workers, suggesting the reproductive bias of the piRNA expression [18]. Watson et al. further surveyed the expression levels of piRNAs in various *A. mellifera* reproductive tissues such as the ovaries, spermatheca, semen, fertilized eggs, unfertilized eggs, and testes [19]. Recently, on the basis of transcriptome data and bioinformatics, our team identified 843 piRNAs in the larval gut of *Apis mellifera ligustica* and uncovered the potential roles of DEpiRNAs during the developmental process of the larval gut [20]. However, little progress for piRNAs in *A. cerana*, the sister species of *A. mellifera,* has been made until now.

To decipher the differential expression pattern of piRNA during the developmental process of *A. c. cerana* larval guts and the regulatory roles of DEpiRNAs, based on our previously gained high-quality small RNA-seq datasets. Identification and structural analysis of piRNAs in *A. c. cerana* larval guts were conducted, followed by investigation of DEpiRNAs and discussion of their potential roles in the regulation of larval gut development. To our knowledge, this is the first documentation of *A. cerana* piRNAs. The findings in the present study will enrich the reservoir of *A. cerana* piRNAs and deepen our understanding of ncRNA-modulated development of larval guts. 

## 2. Materials and Methods

### 2.1. Transcriptome Data Source

The *A. c. cerana* larvae that were used in this study were obtained from colonies reared in the apiary at the College of Animal Sciences (College of Bee Science), Fujian Agriculture and Forestry University, Fuzhou City, China. In a previous study, 4-, 5-, and 6-day-old (Ac4, Ac5, and Ac6 groups) larval gut tissues were respectively prepared and subjected to RNA isolation, cDNA library construction, and deep sequencing using sRNA-seq technology; there were three biological replicas of each group, and each group contained three larval guts [14]. In total, 11,273,306; 11,349,964; and 11,122,092 raw reads were generated from the Ac4, Ac5, and Ac6 groups, respectively. Additionally, 9,791,926; 9,402,531; and 9,394,648 clean tags were respectively gained after quality control [14]. The raw data that were generated from sRNA-seq were deposited in the NCBI SRA database and linked to the BioProject number PRJNA395108.

### 2.2. Identification and Analysis of piRNAs

We followed the previously described method that was used by Xu et al. [20]: (1) the clean reads from each group were aligned to the *A. cerana* reference genome (Assembly ACSNU-2.0) to obtain mapped reads, and then, other types of sRNAs were filtered. (2) small ncRNAs, such as rRNA, scRNA, snoRNA, snRNA, and tRNA, were filtered out by mapping the remaining clean reads to the GenBank and Rfam (11.0) databases. (3) miRNAs in the remaining clean reads were further filtered out. (4) sRNAs between 24 nt and 33 nt in length were retained according to the length characteristics of the piRNAs, and finally only sRNAs mapped to a unique position were retained as candidate piRNAs. 

The expression of each piRNA was normalized using the TPM method (TPM = T × 10^6^/N, where T stands for clean reads of piRNA and N stands for clean reads of total sRNAs). Subsequently, the structural features of the piRNAs, including the length distribution and first base bias, were analyzed based on the prediction result. An UpSet plot and Venn analysis of the piRNA expression levels were visualized using the OmicShare platform (https://www.omicshare.com/tools/, (accessed on 29 November 2022)).

### 2.3. Investigation and Target Prediction of DEpiRNAs

Following the criteria of |log_2_ fold change| ≥ 1 and *p* ≤ 0.05, DEpiRNAs in the Ac4 vs. Ac5 and Ac5 vs. Ac6 comparison groups were screened out. The target mRNAs of DEpiRNA were then predicted using the TargetFinder software [21], the piRNA sequences were compared to the genomic sequences, and only those that were precisely matched and complementary to each other were retained (three mismatches were allowed). Then, each sequence that could be targeted by the piRNAs was scored to predict the piRNA target loci and target mRNAs. Next, using the BLAST tool, the target mRNAs were respectively aligned to the GO (https://www.geneontology.org, (accessed on 29 November 2022)) and KEGG (https://www.genome.jp/kegg/, (accessed on 29 November 2022)) databases to gain functional and pathway annotations. 

### 2.4. Analysis of DEpiRNA-mRNA Regulatory Network

Based on the predicted targeting relationships, regulatory networks between DEpiRNA and target mRNAs were constructed following the thresholds of free energy < −20 kcal mol^−1^ and *p* < 0.05, followed by visualization using the Cytoscape software. Further, on the basis of the KEGG pathway annotations, the target mRNAs annotated in Wnt, TGF-β, and Hippo signaling pathways were further surveyed to construct corresponding regulatory networks, which were then visualized with the Cytoscape software v.3.3.0 [22].

### 2.5. Stem-Loop RT-PCR of piRNAs

The total RNA from 4-, 5-, and 6-day-old *A. c. cerana* larval guts were extracted using a FastPure^®^ Cell/Tissue Total RNA Isolation Kit V2 (Vazyme, Nanjing, China). The concentration and purity of the RNA were checked with a Nanodrop 2000 spectrophotometer (Thermo Fisher, Waltham, MA, USA). A total of three DEpiRNAs (piR-ace-1010100, piR-ace-1183555, and piR-ace-202265,) from the Ac4 vs. Ac5 comparison group were randomly selected for stem-loop RT-PCR validation, and three (piR-ace-828146, piR-ace-904144, and piR-ace-11093) from Ac5 vs. Ac6 comparison group. Specific stem-loop primers and forward primers (F) as well as universal reverse primers (R) were designed using DNAMAN software and then synthesized by Sangon Biotech (Sangon Biotech, Shanghai, China). According to the instructions of HiScript^®^ 1st Strand cDNA Synthesis Kit, cDNA was synthesized by reverse transcription using stem-loop primers and used as templates for PCR of DEpiRNA. Reverse transcription was performed using a mixture of specific loop primers and oligo (dT) primers, and the resulting cDNA was used as templates for PCR. The PCR system (20 μL) contained 1 μL of diluted cDNA, 10 μL of PCR mix (Vazyme, Nanjing, China), 1 μL of forward primers, 1 μL of reverse primers, and 7 μL of diethyl pyrocarbonate (DEPC) water. The PCR was conducted on a T100 thermocycler (Bio-Rad, Hercules, CA, USA) under the following conditions: pre-denaturation step at 95 °C for 5 min; 40 amplification cycles of denaturation at 95 °C for 10 s, annealing at 55 °C for 30 s, and elongation at 72 °C for 1 min, followed by a final elongation step at 72 °C for 10 min. The amplification products were detected on 1.8% agarose gel electrophoresis with Genecolor (Gene-Bio, Shenzhen, China) staining.

### 2.6. RT-qPCR Detection of DEpiRNAs

The RT-qPCR process was carried out following the protocol for the SYBR Green Dye kit (Vazyme, Nanjing, China). The reaction system (20 μL) included 1 μL of cDNA, 1 μL of forward primers, 1 μL of reverse primers, 7 μL of DEPC water, and 10 μL of SYBR SYBR Green Dye. The RT-qPCR process was conducted on an Applied Biosystems QuantStudio 3 system (Thermo Fisher, Waltham, MA, USA) under the following conditions: a pre-denaturation step at 95 °C for 5 min, 40 amplification cycles of denaturation at 95 °C for 10 s, annealing at 60 °C for 30 s, and elongation at 72 °C for 15 s, followed by a final elongation step at 72 °C for 10 min. The reaction was performed using an Applied Biosystems QuantStudio 3 Real-Time PCR System (Thermo Fisher, Waltham, MA, USA). Here, snRNA U6 was selected as the inner reference gene. All the reactions were performed in triplicate. The relative expression of the piRNAs was calculated using the 2^−ΔΔCt^ method [23]. Detailed information about the primers that were used in this work is shown in Appendix A.

### 2.7. Statistical Analysis

The statistical analyses were conducted with SPSS (IBM, Amunque, NY, USA) and GraphPad Prism 7.0 software programs (GraphPad, San Diego, CA, USA). The data are presented as the means ± the standard deviation (SD). The statistical analysis was performed using Student’s *t* test. The significant (*p* < 0.05) GO terms and KEGG pathways were filtered by performing Fisher’s exact test with R software 3.3.1 [24,25].

## 3. Results

### 3.1. Identification, Structural Analysis, and Validation of piRNAs in A. c. cerana Larval Guts

In total, 621, 558, and 549 piRNAs were identified in the Ac4, Ac5, and Ac6 groups, respectively. In addition, 499 piRNAs were shared by the aforementioned three groups, whereas the quantities of specific ones equaled 79, 37, and 11, respectively (Figure 1).

The structural analysis indicated that the length distribution of the identified *A. c. cerana* piRNAs in the abovementioned three groups was from 24 nt to 33 nt (Figure 2A–C); additionally, the first bases of the piRNAs in the Ac4, Ac5, and Ac6 groups had a C bias (Figure 2E,F).

### 3.2. Differential Expression Pattern of piRNAs during the Developmental Process of A. c. cerana Larval Guts

A total of ten DEpiRNAs were discovered in the Ac4 vs. Ac5 comparison group, including six up-regulated and four down-regulated ones. Among these, the most significantly up-regulated three piRNAs were piR-ace-1183555 (log_2_FC = 1.816, *p* = 0.001), piR-ace-458012 (log_2_FC = 1.469, *p* = 0.003), and piR-ace-512574 (log_2_FC = 1.469, *p* = 0.003), while the three most significantly down-regulated ones were piR-ace-1103846 (log_2_FC = −12.019, *p* = 5.32 × 10^−6^), piR-ace-762269 (log_2_FC = −1.251, *p* = 0.043), and piR-ace-1010100 (log_2_FC = −1.228, *p* = 0.048) (Figure 3A). Comparatively, 22 up-regulated DEpiRNAs were characterized in the Ac5 vs. Ac6 comparison group, whereas no down-regulated ones were detected. The most significantly up-regulated DEpiRNA was piR-ace-750627 (log_2_FC = 1.145, *p* = 0.002), followed by piR-ace-11093 (log_2_FC = 1.666, *p* = 0.006) and piR-ace-748816 (log_2_FC = 1.084, *p* = 0.006) (Figure 3B). In addition, the seven DEpiRNAs that were specific for the Ac4 vs. Ac5 comparison group included three up-regulated and four down-regulated piRNAs, while 19 up-regulated DEpiRNAs were specific for the Ac5 vs. Ac6 comparison group (Figure 3C). See Appendix A for the detailed information.

### 3.3. Analysis and Annotation of DEpiRNA-Targeted Genes

In the Ac4 vs. Ac5 comparison group, nine DEpiRNAs were found to target 9011 mRNAs, which could be annotated to 17 biological process-associated GO terms such as biological adhesion and biological regulation, ten molecular function-associated terms such as binding and transcription factor activity and protein binding, and 14 cellular component-associated terms such as membrane part and membrane (Figure 4A). 

In comparison, 21 DEpiRNAs in the Ac5 vs. Ac6 comparison group were detected to target 28,969 mRNAs, which could be annotated to 20 function terms relative to biological processes such as cellular process and localization, ten terms that were relevant to molecular functions such as transporter activity and molecular function regulator, and 16 terms that were related to cellular components such as synapse part and synapse. (Figure 4B).

Additionally, the target genes of DEpiRNAs in the Ac4 vs. Ac5 comparison group could be annotated to 137 KEGG pathways that were associated with environmental information processing, metabolism, organismal systems, genetic information processing, and human diseases, as well as cellular processes, such as endocytosis, fatty acid biosynthesis, and the Wnt signaling pathway (Figure 5A). Comparatively, the DEpiRNA-targeted mRNAs in the Ac5 vs. Ac6 comparison group could be annotated to 164 pathways, including the AGE-RAGE signaling pathway in diabetic complications, ubiquitin-mediated proteolysis, and the TGF-beta signaling pathway (Figure 5B).

### 3.4. Regulatory Network between DEpiRNAs and Target Genes

In the Ac4 vs. Ac5 comparison group, each DEpiRNA can target multiple mRNAs, with piR-ace-748815 and piR-ace-512574 binding to the highest number of targets (1701 and 1718). Similarly, each DEpiRNA in the Ac5 vs. Ac6 comparison group can target several mRNAs, with piR-ace-716466 and piR-ace-828146 linking to the highest number of targets (2089 and 2620). 

As shown in regulatory network, there were complex regulatory relationships between DEpiRNAs and target mRNAs in the aforementioned two comparison groups. A further analysis demonstrated that 105 and 178 target mRNAs in the above-mentioned two comparison groups were involved in the Wnt signaling pathway, Jak/STAT pathway, and TGF-β signaling pathway (Figure 6). Detailed information about DEpiRNAs and corresponding target mRNAs are presented in Appendix A. 

### 3.5. Verification of DEpiRNAs by Stem-Loop RT-PCR and RT-qPCR

A total of six randomly selected DEpiRNAs were subjected to stem-loop RT-PCR verification, and the result was indicative of their expression in the larval gut (Figure 7).

Further, the RT-qPCR result suggested that the expression trends of these six DEpiRNAs were consistent with those in the transcriptome datasets, confirming the authenticity and reliability of our sequencing data (Figure 8).

## 4. Discussion

The previous studies on honeybee piRNAs were mainly associated with *A. mellifera*. Here, on the basis of our previously gained sRNA-seq data, we identified 621, 558, and 549 piRNAs in 4-, 5-, and 6-day-old *A. c. cerana* larval guts, respectively, for the first time. Following the established method [26,27], a total of 621 *A. c. cerana* piRNAs were obtained after removing redundant ones, which will offer a valuable resource for further functional investigations and mechanism exploration studies in the future. In addition, the length distribution of the identified *A. c. cerana* piRNAs was in the range of 24~33 nt, which was consistent with those piRNAs that were identified in other insects and mammals, such as *B. mori* and humans [3]. The gonad tissues were considered to be the primary generative location of piRNAs. However, accumulating evidence has indicated that piRNAs are also abundantly expressed in other organs and tissues of animals [26]. Morazzani et al. revealed that piRNAs were widespread and abundant in the mosquito soma via next-generation sequencing [27]. Feng et al. identified 3396 piRNAs in the midgut tissues of *B. mori* using Illumina sequencing and bioinformatics [26]. We observed that the overall expression levels of the identified piRNAs in *A. c. cerana* larval gut displayed an up-regulation–down-regulation trend (Appendix A), which was indicative of the involvement of piRNAs in the development of the *A. c. cerana* larval gut. Comparatively, although more piRNAs were identified in *A. m. ligustica*, the overall expression levels of piRNAs in the guts of *A. m. ligustica* 4-, 5-, and 6-day-old larvae were more consistent during development [20]. Together, these results demonstrated that the piRNAs had different dynamics during the developmental process of the larval gut of the abovementioned two different bee species, implying different piRNA-regulated mechanisms underlying larval gut development. 

An array of studies suggested that piRNAs can exert regulatory functions by targeting mRNAs via base pairing in various species, such as *Drosophila* [28], *C. elegans* [29], and mice [30]. In silkworms, piRNAs were reported to be involved in sex determination through the down-regulation of target genes [31]. In addition, it is proposed that piRNAs could also interact with targets without causing cleavage following perfect base pairing [29]. A recent study in *C. elegans* defined the piRNA seed region from the 2nd to 7th nt and observed that base pairing outside of the seed region contributed to piRNA target recognition [29]. Here, on the basis of bioinformatics, 9 and 21 piRNAs in Ac4 vs. Ac5 and Ac5 vs. Ac6 comparison groups were predicted to respectively target 9011 and 28,969 mRNAs; among these, piR-ace-748815 and piR-ace-1183555 were found to bind to the highest numbers of targets, followed by piR-ace-716466 and piR-ace-828146, indicative of the higher connectivity of these DEpiRNAs, which deserves additional investigation.

The Wnt signaling pathway performs essential functions in a number of biological processes, from embryogenesis and adult homeostasis to the regulation of cell proliferation, cell polarity, and the specification of cell fate [32]. In insects, the Wnt signaling pathway was found to exert regulatory functions in developmental processes. Oberhofer et al. reported that the Wnt signaling pathway activates germ-layer-related genes, including the pair-rule, Tc-caudal, and Tc-twist genes, and further acts on growth zone metabolism and cell division, which indicated that the Wnt pathway is required for hindgut development [33]. Here, it was detected that mRNAs that were relevant to the Wnt signaling pathway were targeted by six DEpiRNAs (piR-ace-1183555, piR-ace-512574, piR-ace-202265, piR-ace-748815, piR-ace-1103846, and piR-ace-252931) in the Ac4 vs. Ac5 comparison group and 21 DEpiRNA (piR-ace-11093, piR-ace-24995, piR-ace-748814, etc.) in the Ac5 vs. Ac6 comparison group, which suggested that these DEpiRNAs were potentially engaged in regulating the development of the *A. c. cerana* larval gut via the Wnt signaling pathway. More efforts are needed to decipher the underlying mechanism.

The signaling pathways are highly interconnected and extremely diverse in the regulation of cellular communication, which is fundamental to all organisms and mediates numerous processes, such as cell fate decisions, proliferation, migration, and homeostasis [34]. In insects, a subseries of pathways has been identified to date, including the Notch, Wnt, Hedgehog, TGF-β, Hippo, NF-*κ*B, and JAK/STAT signaling pathways [34]. The TGF-β signaling pathway is highly correlated with the expression of ncRNAs such as miRNAs and lncRNAs, TGF-β indirectly represses miR-200 family members through the induction of ZEB1 and ZEB2 transcription factors, and miR-200 family members target and inhibit the expression of ZEB proteins, which stabilize the regulatory networks for TGF-β-induced epithelial–mesenchymal transition [35,36]. In the present study, we observed that TGF-β signaling pathway-related mRNAs were targeted by four DEpiRNAs (piR-ace-748815, piR-ace-1183555, piR-ace-458012, and piR-ace-512574) in the Ac4 vs. Ac5 comparison group and 18 DEpiRNAs (piR-ace-1188284, piR-ace-934193, piR-ace-1183075, etc.) in the Ac5 vs. Ac6 comparison group. This indicated that the aforementioned DEpiRNAs were likely to participate in controlling the TGF-β signaling pathway by modulating downstream gene expression, further regulating larval gut development.

As an important downstream mediator for a variety of eukaryotes, the Jak-STAT signaling pathway plays an essential role in metabolism. Dodington et al. found that JAK-STAT signaling in the peripheral metabolic organs has been shown to regulate a multitude of metabolic processes with the use of tissue-specific knock-out mice [37]. In the *Drosophila* gut, JAK/STAT signaling is essential for intestinal stem cell differentiation and for intestinal regeneration after damage and infection [38]. Here, we found that JAK-STAT signaling pathway-related genes were targeted by five DEpiRNAs (piR-ace-748815, piR-ace-1183555, piR-ace-1103846, piR-ace-512574, and piR-ace-202265) in the Ac4 vs. Ac5 comparison group and 18 DEpiRNAs (piR-ace-1188284, piR-ace-934193, piR-ace-1183075, etc.) in the Ac5 vs. Ac6 comparison group. This indicated that the aforementioned DEpiRNAs have a high correlation with the Jak-STAT signaling pathway and are likely to participate in the metabolism process of the midgut during *A. c. cerana* larval development.

Previously, the overexpression and knockdown of piRNAs by feeding or injection were verified to be effective in several animals, such as *Culex pipiens pallens*, *Nematostella vectensis*, and *Bemisia tabaci* [39,40,41]. In the near future, we will conduct a functional investigation of candidate DEpiRNAs that were identified in this work through overexpression and knockdown.

## 5. Conclusions

Taken together, 621 piRNAs were identified for the first time in the *A. c. cerana* larval gut. The overall expression pattern changes for piRNAs during the developmental process of the larval gut were identified. DEpiRNAs potentially modulate the development of the *A. c. cerana* larval gut by regulating an array of functional terms and pathways, such as the Wnt, TGF-β, and Jak-STAT signaling pathways.

## Figures and Tables

**Figure 1 insects-14-00016-f001:**
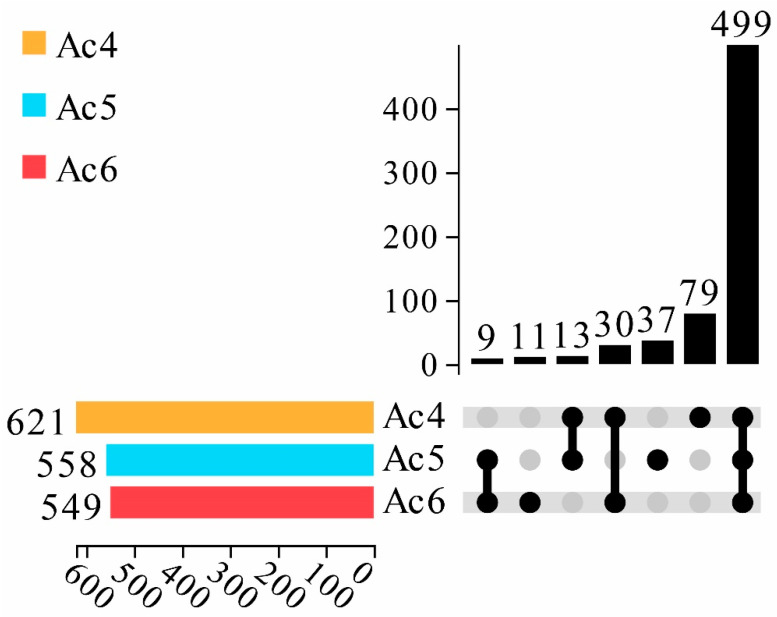
Upset plot of differential piRNA expression in *A. c. cerana* worker larval mid gut. The lower left color column indicates the number of piRNA contained in different groups; the lower right node indicates the piRNA common to each group; the connection between each group indicates the common piRNA; the teamless node indicates the unique piRNA of this group; and the number of unique and common piRNA are presented above; they are arranged in ascending order from left to right.

**Figure 2 insects-14-00016-f002:**
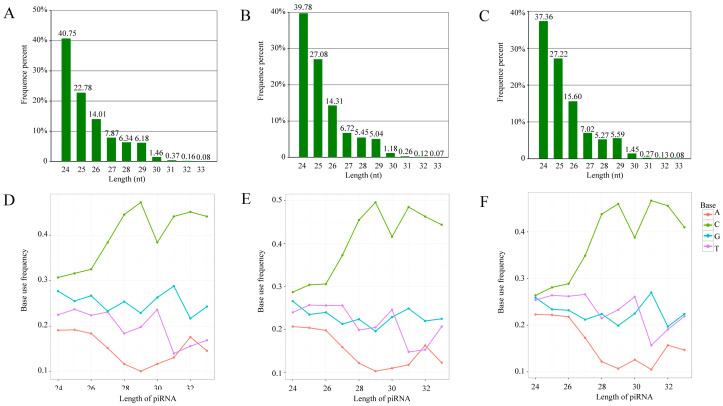
Length distribution and first base bias of *A. c. cerana* piRNAs. (**A**–**C**) Length distribution of piRNAs identified in Ac4, Ac5, and Ac6 groups. The number above each column indicates the percentage of piRNAs distributed at the specific length. (**D**–**F**) First base bias of piRNAs that were identified in Ac4, Ac5, and Ac6 groups. Each broken line represents the frequency of base use at the length distribution ranging from 24 nt to 32 nt.

**Figure 3 insects-14-00016-f003:**
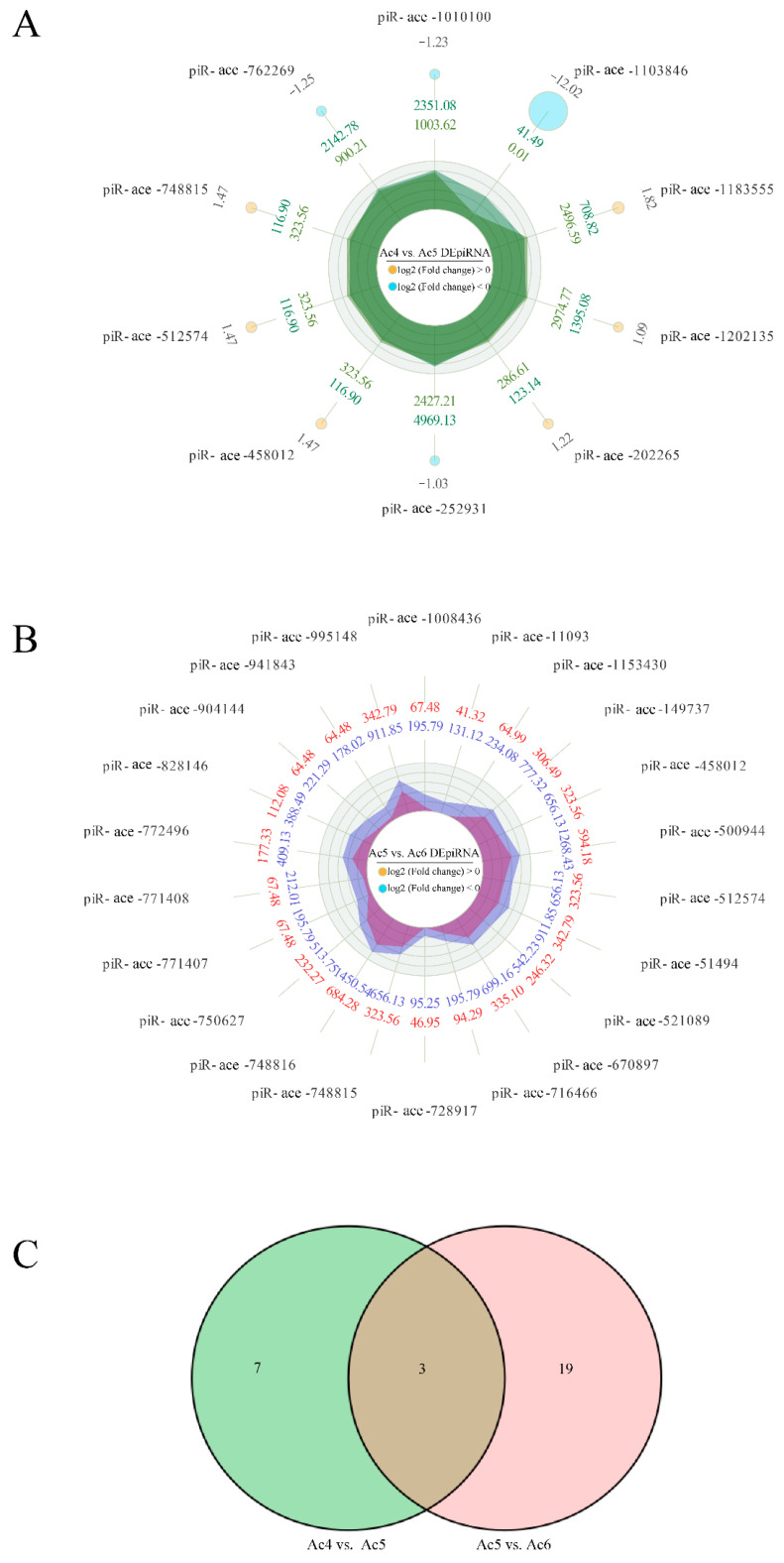
Radar maps and Venn diagram of DEpiRNAs. (**A**) Radar map of DEpiRNAs in the Ac4 vs. Ac5 comparison group. (**B**) Radar map of DEpiRNAs in the Ac5 vs. Ac6 comparison group. (**C**) Venn analysis of DEpiRNAs in Ac4 vs. Ac5 and Ac5 vs. Ac6 comparison groups. There were three up-regulated piRNAs that were shared by the above-mentioned two groups.

**Figure 4 insects-14-00016-f004:**
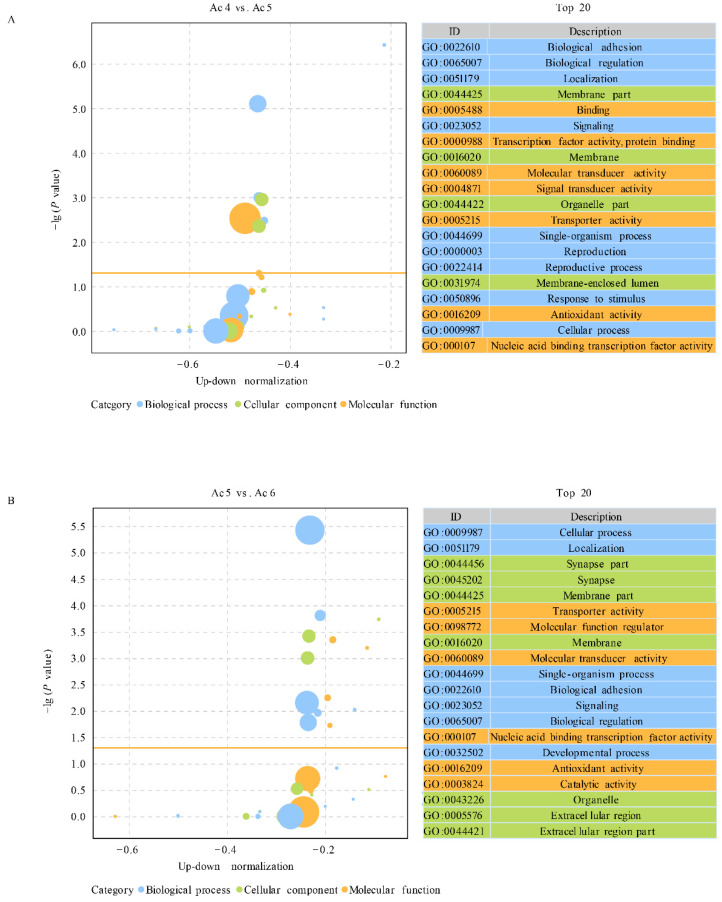
GO terms annotated by target mRNAs of DEpiRNAs. (**A**) Bubble diagram of DEpiRNA-targeted mRNAs in Ac4 vs. Ac5 comparison group. (**B**) Bubble diagram of DEpiRNA-targeted mRNAs in Ac5 vs. Ac6 comparison group.

**Figure 5 insects-14-00016-f005:**
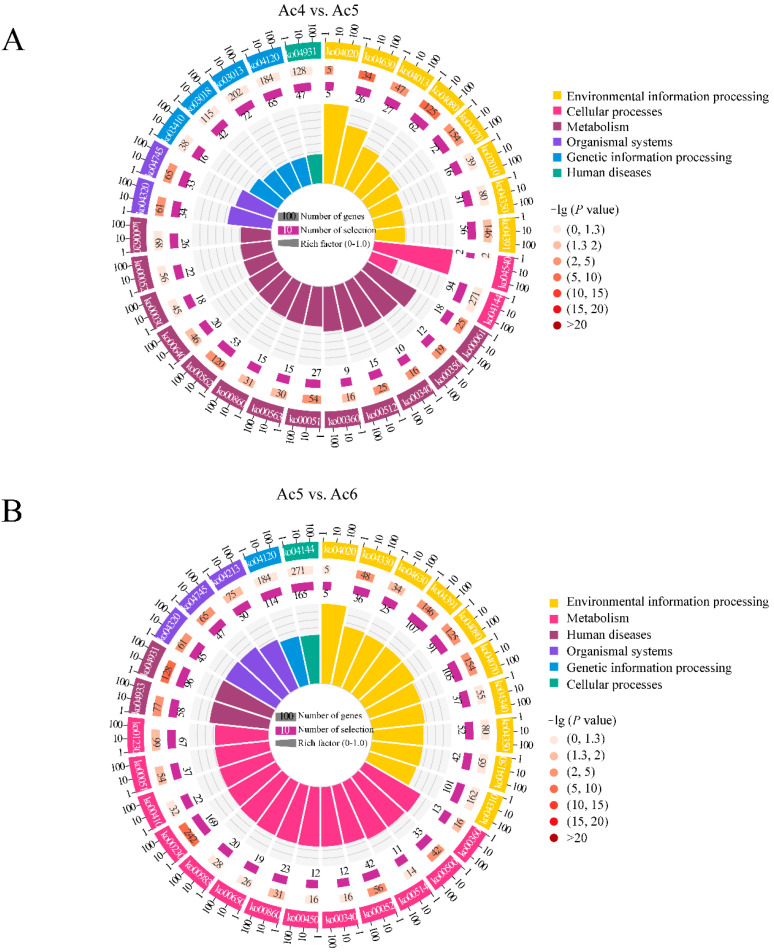
KEGG pathways annotated by target mRNAs of DEpiRNAs. (**A**) Pathways annotated by DEpiRNA-targeted mRNAs in the Ac4 vs. Ac5 comparison group; (**B**) Pathways annotated by DEpiRNA-targeted mRNAs in the Ac5 vs. Ac6 comparison group. From outside to inside, the first circles indicate the enriched terms. The second circles indicate the numbers of targets that were enriched in this term in the background gene as well as the *p*-values, the redder the color, the larger the gene number and the smaller the *p*-value. The third circles indicate the bar charts of the proportion of up-regulated or down-regulated targets, dark purple represents the proportion of up-regulated targets, while light purple represents the proportion of down-regulated targets. The fourth circles indicate the setting standard for the numbers of targets, different colors represent different terms. Rich factor means the number of foreground targets that were enriched in this term divided by the number of background genes.

**Figure 6 insects-14-00016-f006:**
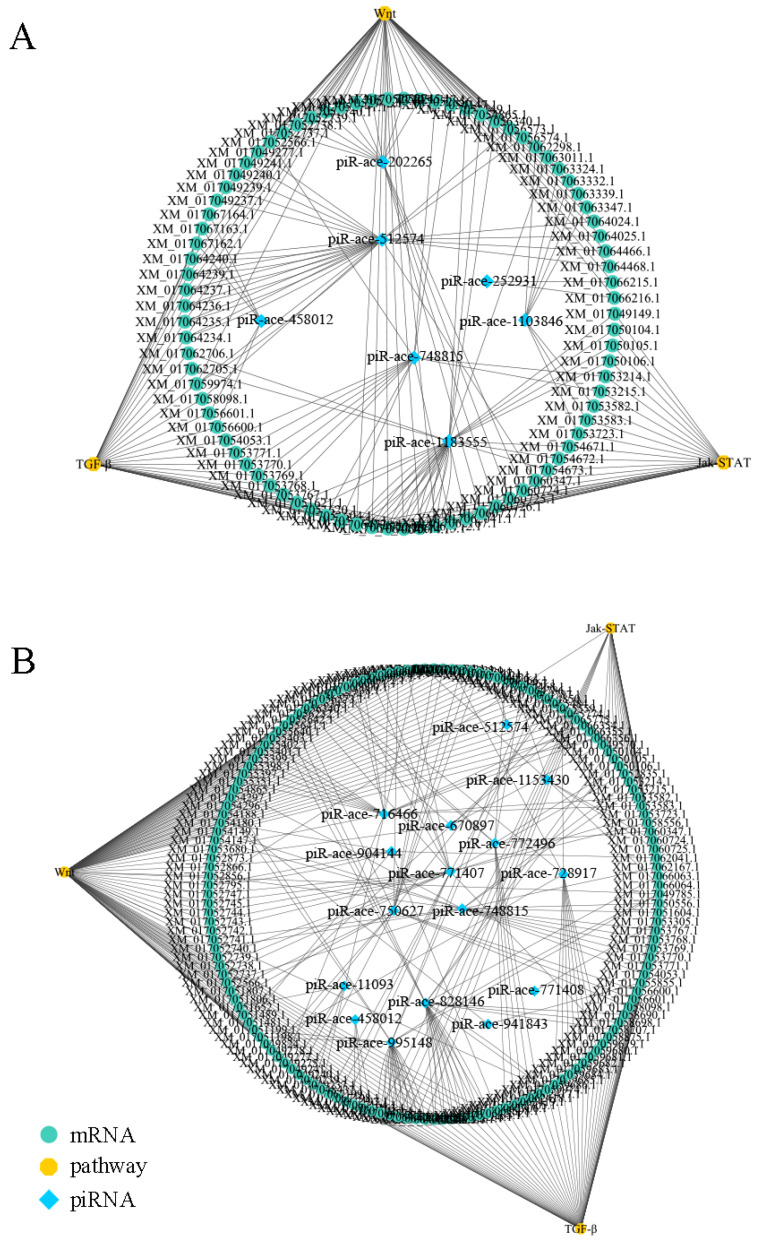
Regulatory network of *A. c. cerana* worker larvae. (**A**) Regulatory network of DEpiRNAs in Ac4 vs. Ac5. (**B**) Regulatory network of DEpiRNAs in Ac5 vs. Ac6.

**Figure 7 insects-14-00016-f007:**
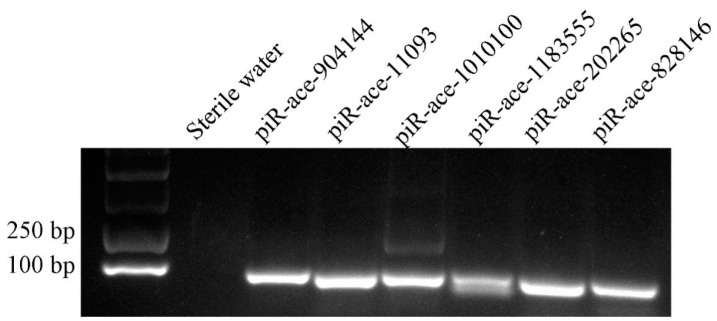
Stem-loop RT-PCR validation of six *A. c. cerana* DEpiRNAs.

**Figure 8 insects-14-00016-f008:**
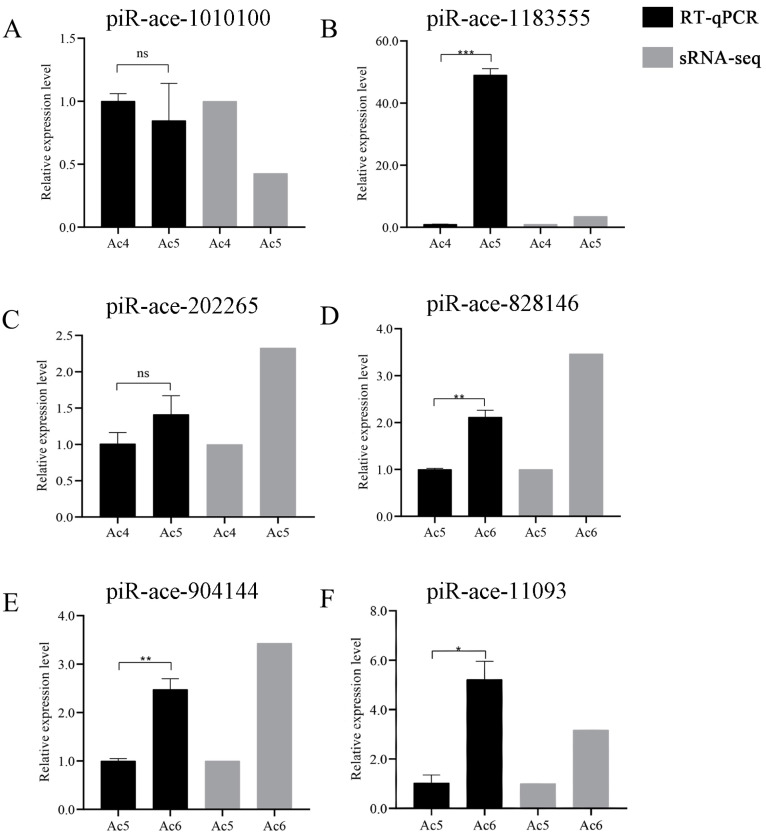
RT-qPCR verification of six *A. c. cerana* DEpiRNAs. (**A**–**C**) Relative expression levels of piRNAs from the Ac4 vs. Ac5 comparison group. (**D**–**F**) Relative expression level of piRNA from the Ac5 vs. Ac6 comparison group. ns indicates non-significant, * indicates *p* < 0.05, ** indicates *p* < 0.01 and *** indicates *p* < 0.001.

## Data Availability

The data presented in this study are openly available in the NCBI SRA database and linked to the BioProject number: PRJNA395108.

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
