# Peer review of "First Identification and Investigation of piRNAs in the Larval Gut of the Asian Honeybee, Apis cerana"

_insects, 2022, doi:10.3390/insects14010016_

Round 1

Reviewer 2 Report

Greetings and Regards

The main research question is First identification and investigation of piRNAs in the larval guts of the Asian honey bee, Apis cerana.

The introduction is correctly described, I checked, but overall I am not convinced that this is the first report.

And due to the importance of investigating and identifying different species of bees, this article is suitable for publication in the insect journal.

The results of the article, according to the authors, not only new insight into the development of A.c. cerana larval gut but also provide a basis for discovering the epigenetic mechanism underlying larval gut development. And in general, it is very important in the world of entomology.

 It is suggested to compare the species in other countries in the future.

Figures 2 and 3 are not clear and need further explanation.

In terms of written language, it needs some changes.

In general, the article is suitable for publication with a few changes

Reviewer 3 Report

Summary

This manuscript performs a computational analysis for piRNAs identification, regulatory network scanning, and validate through RT-PCR using 4-, 5-, and 6-day-old A. cerana larval guts. The authors rely on a search-filter method to define the piRNAs list in Asian honey bee. It gives a length distribution that peaks at 24nt and gradually decrease till 33nt. However, piRNAs has peak around ~30nt based on previous study. This inconsistency suggests that the computational pipeline neglects some important fraction of RNA classes (e.g. circle RNAs). Given that the limited gene annotation of Asian honey bee, it is not a robust method to obtain a list of piRNAs through search-filter.

Major comments

1.     As mentioned above, I have a big concern on the computational pipeline to identify the piRNAs in Asian honeybee.

Minor comments

There are grammars problem. Some examples:

Ln 25, “piRNAs each group were ranged from 24 nt to 33 nt in length” should be “piRNAs in each group ranged from 25 24 nt to 33 nt in length”

Ln 82 “fertilised” be ““fertilized”

Ln 355 “we will conduct functional investigation” miss “conduct a functional investigation”

Inconsistency “A. cerana”, “ Apis cerana cerana”, “ A. c. cerana”

I suggest to improve the writing.

Round 2

Reviewer 3 Report

Overall, the authors' response address my minor concerns.

The authors did not answer my major concern about the peak length of piRNAs should be around 29 or 30nt, as shown in previous publication. All the reference paper provide the similar profile range from 18 to 32nt and peaking at ~30nt.

One possible explanation could be the published piRNA identified from piwi-complex. Due to the insufficient annotation of honey bee, the pools still contain some other small RNAs. So a screen solely rely on computational pipeline give a mix of piRNAs and others.

The final list is still useful.